Prognostic impact of serum interleukin-6 and 17 level in patients with bladder cancer: a systematic review and meta-analysis

http://orcid.org/0000-0003-1372-4596 Liu Liang 1 2 3 mnwkll@163.com
He Jun-Hui 4
Xiao Yu 5
Wei Dong 6
1 Department of Urology, Baoding No.1 Central Hospital , Baoding, Hebei , China
2 Key Laboratory of Molecular Pathology and Early Diagnosis of Tumor in Hebei Province , Baoding, Hebei , China
3 Prostate and Andrology Key Laboratory of Baoding , Baoding, Hebei , China
4 Department of Urology, Heze Municipal Hospital , Heze, Shandong , China
5 Psychosomatic Medical Center, The Fourth People’s Hospital of Chengdu , Chengdu, Sichuan , China
6 Department of Urology, Hebei General Hospital , Shijiazhuang, Hebei , China
Ozdag Sevgili Hilal
Electronic publication date: 2025 Apr 29
Publication date: 2025
Volume: 13
Electronic Location ID: e19385
Received 2024 Oct 28; Accepted 2025 Apr 7
Copyright: © 2025 Liu et al.
Copyright year: 2025
Copyright holder: Liu et al.
License: This is an open access article distributed under the terms of the Creative Commons Attribution License, which permits unrestricted use, distribution, reproduction and adaptation in any medium and for any purpose provided that it is properly attributed. For attribution, the original author(s), title, publication source (PeerJ) and either DOI or URL of the article must be cited.
License URL: https://creativecommons.org/licenses/by/4.0/

Keywords: Urinary bladder neoplasms, Bladder cancer, Interleukin-6, Interleukin-17, Meta-analysis, Systematic review

Funding: Medical Science Research Project of Hebei 20241833 and 20251438 Science and Technology Bureau of Baoding Province 2341ZF103 This study was supported by grants from the Medical Science Research Project of Hebei (Item No: 20241833, 20251438), and the Science and Technology Bureau of Baoding (Item No: 2341ZF103). The funders had no role in study design, data collection and analysis, decision to publish, or preparation of the manuscript.

==============================
Background

Interleukin, a noninvasive biomarker, holds huge potential for providing valuable insights into the management of inflammatory conditions and tumor diseases. This systematic review and meta-analysis aimed to evaluate the prognostic role of interleukin in bladder cancer (BCa) patients.

Methods

A comprehensive search of six English and Chinese databases (PubMed, Embase, Web of Science, Chinese National Knowledge Infrastructure (CNKI), Wanfang, and Chinese Biomedical Literature Service System) was conducted from inception to July 10, 2024. Studies investigating the association between serum interleukin levels and BCa were included. Outcome measures encompassed disease-free survival (DFS), overall survival (OS), and disease-specific survival (DSS). Statistical analyses were performed using RevMan 5.4.1, employing random or fixed-effects models as appropriate. Sensitivity, subgroup, and descriptive analyses were also conducted.

Results

A total of seven studies involving 1,505 patients were included. Four studies reported the association between serum interleukin-6/17 (IL-6/17) and OS in BCa. Patients with elevated serum IL levels exhibited a worse OS (HR = 2.28; 95% CI [1.03–5.05]; P = 0.04); however, subgroup analysis revealed that only high serum IL-17 levels were significantly associated with shorter OS, while IL-6 levels showed no association with OS. Six studies examined the relationship between serum IL-6/17 and DFS in BCa. Patients with elevated serum IL levels were associated with poorer DFS (HR = 2.57; 95% CI [1.55–4.26]; P < 0.001). This association remained consistent across subgroup analyses based on interleukin type, publication country, and surgical methods. Only two studies investigated the relationship between serum IL-6/17 and DSS in BCa, with no significant association found (HR = 1.58; 95% CI [1.00-2.51]; P = 0.05).

Conclusion

This meta-analysis demonstrates a strong association between serum interleukin levels and survival outcomes in BCa, suggesting that serum interleukin testing may be a valuable clinical tool for predicting patient outcomes and guiding treatment decisions.

Introduction

Bladder cancer (BCa) is classified into non-muscle invasive and muscle-invasive subtypes and ranks as the tenth most common cancer globally, characterized by high mortality and recurrence rates (Richters, Aben & Kiemeney, 2019). It is reported that 25% of newly diagnosed BCa cases involve muscle-invasive or metastatic disease (Kamat et al., 2016). The prognostication of BCa has traditionally relied on clinical and pathological evaluations, including TNM staging, molecular subtyping, and assessment of pathological variants. Over the past decade, pathological variations in urothelial carcinoma have received significant attention, especially in diagnostic value and prognostic significance. Prior research (Lobo et al., 2019; Warrick, 2017) has suggested that histologic variants of urothelial carcinoma exhibit more aggressive behavior, display more histologically malignant features, demonstrate poorer treatment response, correlate with shorter survival times, and are associated with higher mortality compared to pure urothelial carcinoma. However, Akbas, Bektas & Yazici (2024) reported no significant differences in overall survival (OS) among histologic variants and identified no prognostic factors for either disease-free or overall survival. Conversely, numerous other studies have reached contrasting conclusions. For instance, Wasco et al. (2007) posited that patients with histologic subtypes and/or divergent differentiation of urothelial carcinoma are more prone to muscularis propria invasion following transurethral resection of bladder tumor (TURBT) and extravesical extension following radical cystectomy (RC), compared to those with pure urothelial carcinoma. These pathological variations also influence treatment strategies and survival outcomes, often necessitating more aggressive interventions such as re-TURBT, cystectomy, or multimodal therapy. A large retrospective study (de Angelis et al., 2024) demonstrated that the median cancer-specific mortality-free survival for patients with organ-confined non-urothelial carcinoma of the urinary bladder treated with trimodal therapy was significantly shorter than that observed in patients with organ-confined urothelial carcinoma of the urinary bladder (36 months vs. 60 months). However, the accuracy of these assessments is significantly influenced by the availability, accessibility, and quality of pathological tissue. Given the minimally invasive nature, low cost, and ease of collection and processing, serum-based examinations offer a promising alternative. Consequently, there is a critical need to identify novel and effective serum biomarkers for BCa that can serve as prognostic indicators.

Cytokines have been implicated in the pathogenesis of bladder urothelial carcinomas. Serum interleukin-6 (IL-6), a pro-inflammatory cytokine and inflammation marker, is dysregulated in multiple cancers, including BCa, prostate cancer, and breast neoplasms (Tong, Hu & Li, 2022). Previous studies in pancreatic cancer have suggested that higher serum IL-6 levels connoted with shorter survival (Mitsunaga et al., 2013; Tsukinaga et al., 2015). Moreover, elevated serum IL-6 levels have been associated with aggressive behavior in urothelial carcinomas of the bladder (Chen et al., 2013; Okamoto, Hattori & Oyasu, 1997). Interleukin-17 (IL-17) has been shown to promote tumor progression by inhibiting apoptosis and stimulating angiogenesis (Wang et al., 2009; Nam et al., 2008). Conversely, IL-17 can enhance antitumor immune responses by regulating T lymphocyte proliferation (Muranski et al., 2008; Kryczek et al., 2009). Additionally, serum IL-17 levels have been utilized to predict lung cancer prognosis (Xu et al., 2014; Zhang et al., 2012). While growing evidence suggests a correlation between serum IL-6 and IL-17 levels and cancer prognosis, the specific impact of these cytokines on BCa patients’ oncologic outcomes remains controversial. Furthermore, the available literature lacks sufficient evidence to draw definitive conclusions. For example, Schuettfort et al. (2021) concluded that elevated IL-6 levels predict shorter OS, recurrence-free survival (RFS), and cancer-specific survival (CSS) in BCa patients. However, Andrews et al. (2002) and Cai et al. (2007) reported that IL-6 levels are not an independent prognostic factor for RFS and CSS.

Therefore, this study sought to assess the prognostic value of serum IL-6 and IL-17 in BCa through a systematic review and meta-analysis of published studies.

Materials and Methods

Protocol and registration

This systematic review and meta-analysis were conducted in accordance with the Preferred Reporting Items for Systematic Reviews and Meta-Analyses (PRISMA), Meta-analysis of Observational Studies in Epidemiology (MOOSE), and Assessing the Methodological Quality of Systematic Reviews (AMSTAR) guidelines (Page et al., 2020; Stroup et al., 2000; Shea et al., 2017). The study was prospectively registered in the International Prospective Register of Systematic Reviews (PROSPERO).

Search strategy

A comprehensive search of six English and Chinese databases—PubMed, Embase, Web of Science, Chinese National Knowledge Infrastructure (CNKI), Wanfang Database, and Chinese Biomedical Literature Service System—was conducted from inception to July 10, 2024. Both subject headings and free-text terms were employed in the search strategy. Search terms include: ‘urinary bladder neoplasms’, ‘urinary bladder neoplasm’, ‘bladder neoplasms’, ‘bladder neoplasm’, ‘bladder tumors’, ‘bladder tumor’, ‘urinary bladder cancer’, ‘bladder cancer’, ‘bladder cancers’, ‘cancer of bladder’, ‘cancer of the bladder’, ‘malignant tumor of urinary bladder’, ‘interleukin-6’, ‘interleukin-17’, ‘IL-6’, ‘IL6’, ‘IL-17’, ‘IL17’, ‘prognosis’, ‘prognoses’, ‘prognostic factors’, and ‘prognostic factor’. Additionally, a manual search of the grey literature was performed. Table 1 outlines the specific search strategy for the PubMed database.

Table 1 PubMed search strategy.

Order	Strategy	
#1	MeSH = Urinary Bladder Neoplasms	
#2	Title/Abstract = Urinary Bladder Neoplasms OR Urinary Bladder Neoplasm OR Bladder Neoplasms OR Bladder Neoplasm OR Bladder Tumors OR Bladder Tumor OR Urinary Bladder Cancer OR Bladder Cancer OR Bladder Cancers OR Cancer of Bladder OR Cancer of the Bladder ORMalignant Tumor of Urinary Bladder	
#3	#1 OR #2	
#4	MeSH = interleukin-6 OR interleukin-17	
#5	Title/Abstract = interleukin-6 OR IL-6 OR IL6 OR interleukin-17 OR IL-17 OR IL17	
#6	#4 OR #5	
#7	MeSH = Prognosis	
#8	Title/Abstract = Prognosis OR Prognoses OR Prognostic Factors OR Prognostic Factor	
#9	#7 OR #8	
#7	#3 AND #6 AND #9	

Outcomes

The primary outcome measures were disease-free survival (DFS) and overall OS. Disease-specific survival (DSS) was a secondary outcome measure. DFS was assessed based on data on disease recurrence survival (DRS), DFS, RFS, and progression-free survival (PFS). DSS was evaluated using data on DSS and CSS.

Inclusion and exclusion criteria

Inclusion criteria were as follows: (1) study population diagnosed with BCa via histopathology and aged ≥18 years; (2) measurement and categorization of serum IL-6 or IL-17 into high and low groups; (3) assessment of at least one of the following outcomes: DFS, DRS, RFS, PFS, OS, DSS, or CSS; (4) study design as an observational study (case-control or cohort); (5) sample size of ≥ 30 participants; (6) provision or calculability of 95% confidence intervals (CIs) for serum IL-6 or IL-17 levels based on presented data; and (7) publication in English or Chinese.

Exclusion criteria were as follows: (1) absence of histological diagnosis, (2) incomplete data on serum IL-6 or IL-17 measurement or outcome, (3) book, patent, thesis, conference abstract, literature review, or case report format, (4) inability to extract data, and (5) duplicate publications.

Literature screening and data extraction

The database searches were conducted by two authors in accordance with the predefined search strategy. Duplicate records were removed using Endnote software and manual methods. Subsequently, two independent researchers screened the remaining citations. Eight studies (Tong, Hu & Li, 2022; Schuettfort et al., 2021; Andrews et al., 2002; Cai et al., 2007; Gong et al., 2019; Liu et al., 2018; Wang et al., 2024; Zhang et al., 2020) were selected for full-text review, of which seven (Tong, Hu & Li, 2022; Schuettfort et al., 2021; Andrews et al., 2002; Gong et al., 2019; Liu et al., 2018; Wang et al., 2024; Zhang et al., 2020) met the inclusion criteria for the meta-analysis. Extracted data included: (1) basic study characteristics (first author, publication year, sample size, sex, age); (2) clinicopathological features (T stage, grade); (3) interleukin-related information (assessment method, interleukin type, test method, cutoff value); (4) treatment and outcome data (surgery type, median follow-up, DFS, DRS, RFS, PFS, OS, DSS, CSS); and (5) meta-analysis data (hazard ratios (HR) and 95% CI). Data extraction and study selection were performed independently by two reviewers, with discrepancies resolved through discussion or, if necessary, by a third reviewer.

Quality assessment

Two investigators independently assessed the quality of the included studies using the Newcastle-Ottawa Scale (NOS) (Stang, 2010). A total score of nine was possible, with studies categorized into three quality levels: low (0–3 points), moderate (4–6 points), and high (7–9 points). Discrepancies between reviewers were resolved through discussion or by involving a third reviewer.

Statistical analysis

Statistical analyses were performed using RevMan 5.4.1. HRs, 95% CIs, and p-values from univariate or multivariate Cox regression were extracted from each study. When relevant data were not reported, Tierney’s method was employed for calculation (Tierney et al., 2007). In instances where both univariate and multivariate analysis results were available for survival outcomes, the latter were prioritized. Heterogeneity was assessed using the I2 statistic (Higgins et al., 2003). A fixed-effects model was used for meta-analysis when I2 was less than 50%, indicating no significant heterogeneity; otherwise, a random-effects model was applied.

Sensitivity analyses were conducted by employing alternative statistical models. To assess the robustness of the meta-analysis results, a sensitivity analysis was performed by sequentially excluding individual studies. Subgroup analyses were conducted to evaluate the prognostic significance of interleukin levels for OS in BCa patients based on interleukin type, and for DFS in BCa patients based on interleukin type, study location, and surgical method. To assess potential heterogeneity among studies, a random-effects model was used to calculate pooled HRs. Publication bias was evaluated using funnel plots when more than seven studies were included. Statistical significance was defined as a p-value < 0.05.

Results

Study characteristics and quality assessment

A total of 292 articles were initially identified through electronic database searches (PubMed (n = 88), Embase (n = 51), Web of Science (n = 66), CNKI (n = 72), Wanfang (n = 2), and SinoMed (n = 12)) and manual searches using the predefined search strategy. After removing duplicates using both software and manual methods, 197 articles remained. Following title and abstract screening, eight articles (Tong, Hu & Li, 2022; Schuettfort et al., 2021; Andrews et al., 2002; Cai et al., 2007; Gong et al., 2019; Liu et al., 2018; Wang et al., 2024; Zhang et al., 2020) were selected for full-text review. Ultimately, one article (Cai et al., 2007) was excluded due to insufficient data quality. The literature screening process is summarized in Fig. 1.

Figure 1 Literature search and study selection flow diagram for systematic reviews.

The NOS scores ranged from 6 to 8 points. Among the evaluated studies (n = 7) (Tong, Hu & Li, 2022; Schuettfort et al., 2021; Andrews et al., 2002; Cai et al., 2007; Gong et al., 2019; Liu et al., 2018; Wang et al., 2024; Zhang et al., 2020), six (Schuettfort et al., 2021; Andrews et al., 2002; Gong et al., 2019; Liu et al., 2018; Wang et al., 2024; Zhang et al., 2020) were deemed high quality, one (Tong, Hu & Li, 2022) was moderate quality, and none were classified as low quality.

Eligible studies and patient characteristics

A total of seven studies (Tong, Hu & Li, 2022; Schuettfort et al., 2021; Andrews et al., 2002; Cai et al., 2007; Gong et al., 2019; Liu et al., 2018; Wang et al., 2024; Zhang et al., 2020) involving 1,505 patients were included in the meta-analysis. One study (Andrews et al., 2002) was published in 2002, while the remaining six (Tong, Hu & Li, 2022; Schuettfort et al., 2021; Gong et al., 2019; Liu et al., 2018; Wang et al., 2024; Zhang et al., 2020) were conducted within the past 5 years, indicating growing interest. Table 2 summarizes the key characteristics of the included studies. Only two (Liu et al., 2018; Zhang et al., 2020) studies investigated the association between serum IL-17 and BCa, whereas the remaining studies focused on the relationship between serum IL-6 and BCa (Tong, Hu & Li, 2022; Andrews et al., 2002; Gong et al., 2019; Schuettfort et al., 2021; Wang et al., 2024). Regarding surgical interventions, patients underwent either transurethral resection of bladder tumor (TURBT) or radical cystectomy (RC).

Table 2 Summary of characteristics.

Author	Year	Size, n	Male/Female	Age, Y	T stage	Grade	Assessment	Test
method	Cut-off	Surgery	Median follow-up (months)	Prognostic condition	HR	95% CI	NOS	
Andrews et al. (2002)	2002	51	47/4	65.0 ± 8.5	Tis/Ta/
T1-4	II-III	Serum
IL-6	ELISA	4.8
pg/mL	RC	45.7	DRS
DSS	DRS:1.417
DSS:2.171	DRS:[0.954–2.107]
DSS:[1.288–3.659]	8	
Gong et al. (2019)	2019	124	100/24	65	T1-4	Low
High	Serum
IL-6	ELISA	NR	RC	NR	OS	Kaplan-Meier	Kaplan-Meier	7	
Liu et al. (2018)	2018	30	16/14	58.3	T0-4	I-III	Serum
IL-17	ELISA	NR	Surgery	30	OS
DFS	Kaplan-Meier	Kaplan-Meier	8	
Schuettfort et al. (2021)	2021	1,036	814/222	66.5	T0/Tis/Ta/T1-4	I-III	Serum
IL-6	ELISA	2.76 pg/mL	RC	37	OS
RFS
CSS	OS:1.2
RFS:1.32
CSS:1.33	OS:[1.13–1.27]
RFS:[1.23–1.41]
CSS:[1.24–1.42]	7	
Tong, Hu & Li (2022)	2022	71	56/15	NR	T1	Low
High	Serum
IL-6	ELISA	4.9 pg/mL	TURBT	NR	RFS	11.27	[3.76–33.80]	6	
Wang et al. (2024)	2024	99	88/11	68.94 ± 11.64	T1	NR	Serum
IL-6	Flow
cytometer	4.150 pg/mL	TURBT	27.71	RFS	NR	[2.737–11.593]	7	
Zhang et al. (2020)	2020	94	65/29	60.19 ± 10.68	T1-4	I-III	Serum
IL-17	ELISA	NR	TURBT	NR	OS
PFS	OS:4.455
PFS:4.39	OS:[1.255-15.818]
PFS:[1.244–15.489]	7	
Note:

Abbreviation: HR, hazard ratio; CI, confidence interval; NR, not reported; IL-6, interleukin-6; IL-17, interleukin-17; ELISA, enzyme linked immunosorbent assay; NOS, Newcastle-Ottawa scale; RC, radical cystectomy; TURBT, transurethral resection of bladder tumour; DRS, disease recurrence survival; DSS, disease specific survival; RFS, recurrence-free survival; CSS, cancer-specific survival; PFS, progression free survival; OS, overall survival; DFS, disease free survival.

Association interleukin levels with survival

Four studies (Schuettfort et al., 2021; Gong et al., 2019; Liu et al., 2018; Zhang et al., 2020) (n = 1,284 patients) reported the association between serum IL-6/17 and OS in BCa. Given the substantial heterogeneity, a random-effects model was employed. Meta-analysis results indicated that patients with elevated serum IL levels exhibited poorer OS (HR = 2.28, 95% CI [1.03–5.05], P = 0.04; Fig. 2A). Due to the limited number of included studies, sensitivity analyses were not performed.

Figure 2 Forest plots demonstrating the association between elevated interleukin levels and oncologic outcomes in patients with BCa. (A) overall survival; (B) disease-free survival; (C) disease-specific survival.

Sources: Andrews et al. (2002), Gong et al. (2019), Liu et al. (2018), Schuettfort et al. (2021), Tong, Hu & Li (2022), Wang et al. (2024), Zhang et al. (2020).

Six studies (Tong, Hu & Li, 2022; Schuettfort et al., 2021; Andrews et al., 2002; Liu et al., 2018; Wang et al., 2024; Zhang et al., 2020) (n = 1,381 patients) reported the association between serum IL-6/17 levels and DFS/DRS/RFS/PFS in BCa. A random-effects model demonstrated that elevated IL levels were associated with poorer DFS/DRS/RFS/PFS (HR = 2.57; 95% CI [1.55–4.26]; P < 0.001; Fig. 2B). Sensitivity analysis demonstrated consistent pooled estimates upon removal of any individual study, supporting the robustness of the meta-analysis.

Only two studies (Schuettfort et al., 2021; Andrews et al., 2002) reported the relationship between serum IL-6/17 and DSS/CSS in BCa. A random-effects model demonstrated no association between elevated serum interleukin levels and DSS/CSS (HR = 1.58; 95% CI [1.00–2.51]; P = 0.05; Fig. 2C). Andrews et al. (2002) analyzed 51 patients who underwent RC and found that serum IL-6 was a predictor of DSS according to their preoperative Cox proportional hazards model (HR = 2.171; 95% CI [1.288–3.659]; P = 0.05). Consistent findings were reported in another study (HR = 1.33; 95% CI [1.24–1.42]; P < 0.001).

Prognostic value of interleukins in subgroup analyses

Subsequent subgroup analyses were performed to evaluate the relationship between serum interleukin subtype and OS. Patients with elevated serum IL-17 levels exhibited significantly shorter OS (HR = 3.32; 95% CI [1.25–8.80]; P = 0.02; Fig. 3). Conversely, serum IL-6 levels were not associated with OS (HR = 1.92; 95% CI [0.67–5.47]; P = 0.22; Fig. 3). However, the pooled analysis revealed an association between elevated serum IL levels and poorer OS (HR = 2.28; 95% CI [1.03–5.05]; P = 0.04; Fig. 3).

Figure 3 Forest plots demonstrating the association between elevated interleukin level and overall survival in patients with BCa for subgroup analysis according to interleukin type.

Sources: Gong et al. (2019), Liu et al. (2018), Schuettfort et al. (2021), Zhang et al. (2020).

Subgroup analyses of DFS/DRS/RFS/PFS were conducted based on serum interleukin type. Four studies (Tong, Hu & Li, 2022; Schuettfort et al., 2021; Andrews et al., 2002; Wang et al., 2024) reported serum IL-6 levels, while two studies (Liu et al., 2018; Zhang et al., 2020) reported serum IL-17 levels. Patients with elevated serum IL-6 exhibited poorer DFS/DRS/RFS/PFS (HR = 2.74; 95% CI [1.37–5.46]; P = 0.004; Fig. 4). Similarly, patients with elevated serum IL-17 levels demonstrated poorer prognosis in BCa (HR = 2.32; 95% CI [1.11–4.84]; P = 0.02; Fig. 4). A stratified analysis based on the country of publication was also undertaken. Elevated serum interleukin levels consistently predicted unfavorable prognosis irrespective of geographic origin: China (HR = 4.42; 95% CI: 1.90-10.30; P < 0.001; Fig. 5) or other countries (HR = 1.32; 95% CI [1.23–1.41]; P < 0.001; Fig. 5). A subgroup analysis stratified by surgical procedure was subsequently performed. Due to the absence of surgical procedure specification in one study (Liu et al., 2018), the analysis was restricted to the remaining five studies (Tong, Hu & Li, 2022; Schuettfort et al., 2021; Andrews et al., 2002; Wang et al., 2024; Zhang et al., 2020). Patients who underwent RC with higher serum interleukin levels exhibited significantly shorter DFS/DRS/RFS/PFS (HR = 1.32; 95% CI [1.23–1.41]; P < 0.001; Fig. 6). In contrast, patients undergoing TURBT and exhibiting elevated serum interleukin levels experienced a significantly worse prognosis (HR = 6.38; 95% CI [3.70–10.99]; P < 0.001; Fig. 6).

Figure 4 Forest plots demonstrating the association between elevated interleukin level and disease-free survival in patients with BCa for subgroup analysis according to interleukin type.

Sources: Andrews et al. (2002), Liu et al. (2018), Schuettfort et al. (2021), Tong, Hu & Li (2022), Wang et al. (2024), Zhang et al. (2020).

Figure 5 Forest plots demonstrating the association between elevated interleukin level and disease-free survival in patients with BCa for subgroup analysis according to country.

Sources: Andrews et al. (2002), Liu et al. (2018), Schuettfort et al. (2021), Tong, Hu & Li (2022), Wang et al. (2024), Zhang et al. (2020).

Figure 6 Forest plots demonstrating the association between elevated interleukin level and disease-free survival in patients with BCa for subgroup analysis according to surgical approach.

Sources: Andrews et al. (2002), Schuettfort et al. (2021), Tong, Hu & Li (2022), Wang et al. (2024), Zhang et al. (2020).

Publication bias

Given the limited number of studies included in this meta-analysis, an assessment of publication bias was not feasible.

Discussion

In recent years, the field of bladder cancer treatment has witnessed significant advancements; however, its therapeutic efficacy remains suboptimal due to the cancer’s high recurrence rate and invasive nature. Early and accurate prognostication of BCa patients is crucial for optimizing clinical management and outcomes. Chronic inflammation has been established as a prevalent feature of various cancers, playing a pivotal role in tumorigenesis and progression. It is considered as a key driver of cancer development (Balkwill & Mantovani, 2001; Abdel-Latif et al., 2009).

To the best of our knowledge, this is the first meta-analysis to investigate the association between IL-6 and IL-17 levels and BCa prognosis. Our meta-analysis revealed several key findings. First, a significant strong association was observed between serum interleukin levels and OS in patients. However, subsequent subgroup analysis indicated that the specific interleukin measured may be a significant source of heterogeneity. Specifically, elevated serum IL-17 levels were predictive of poorer OS in BCa patients, whereas serum IL-6 levels were not associated with OS, a finding that contrasts with previous reports. Additionally, we examined the relationship between IL-6/17 expression and DFS/DSS, observing a significant association between elevated IL-6/17 levels and poorer DFS but not DSS. Further stratification by serum interleukin type, publication country, and surgical method consistently demonstrated a significant association between IL-6/17 expression and DFS.

It is now understood that the low molecular weight of IL-6 enables its free diffusion through intercellular junctions and direct interaction with the tumor microenvironment thereby contributing to tumor cell proliferation. IL-6 can exert its effects on tumor cells through the JAK/STAT3 signaling pathway, with STAT3 activation shown to be crucial in invasive bladder cancer (Ho et al., 2012). Furthermore, IL-6 has been implicated in angiogenesis, immune evasion, and suppression of antitumor T-cell responses (Lee et al., 2008). Additionally, IL-6-induced telomerase activation can prevent or delay telomere-induced senescence and transform non-tumor cells into cancer stem cells (Yamagiwa, Meng & Patel, 2006; Kim et al., 2013). Moreover, IL-6 interacts with growth factor signaling, including epidermal growth factor and hepatocyte growth factor, to regulate tumor cell proliferation (Taniguchi & Karin, 2014). Collectively, these findings suggest a pro-cancerous role for IL-6 in BCa, promoting tumor growth and inhibiting cell apoptosis. However, contradictory results have been reported. Tsui et al. (2013) observed the anti-proliferative, anti-migratory, and anti-invasive effects of IL-6 on BCa cells in vitro, suggesting a potential protective role in BCa patients. Although our subgroup analysis did not reveal an association between IL-6 and OS in BCa patients, the limited number of studies included in this analysis precludes definitive conclusions and necessitates further investigation. Therefore, additional research is required to elucidate the precise prognostic roles and underlying mechanisms of IL-6 in BCa.

IL-17, a proinflammatory cytokine, has been implicated in tumor progression (Doroudchi et al., 2013). Studies have demonstrated that IL-17 can induce tumor cells and fibroblasts to release vascular endothelial growth factors, thereby promoting angiogenesis (Numasaki et al., 2002). IL-17 has also been reported to be highly expressed in ovarian and colorectal cancers, where it contributes to angiogenesis (Straus, 2013). Additionally, IL-17 promotes tumor regeneration and invasiveness in cervical cancer (Punt et al., 2015). Furthermore, IL-17 has been shown to increase active metalloproteinase-9, which enhances angiogenesis and tumor growth (Li & Boussiotis, 2013). Tosolini et al. (2011) reported a worse prognosis in colorectal cancer patients with elevated IL-17 expression, suggesting a pro-oncogenic role for this cytokine. Dowell et al. (2017) observed significantly increased IL-17 levels in patients with bladder tumors of histological grade 3 with carcinoma in situ and linked this elevation to an inflammatory response mediated by IL-6 and IL-8. Collectively, these clinical and experimental high studies, which demonstrate an association between elevated IL-17 levels and poorer prognosis, support the potential of IL-17 as a prognostic biomarker in BCa patients.

Several limitations of this meta-analysis warrant consideration. First, variations in interleukin assay methods across studies may have introduced clinical and statistical heterogeneity. Second, inconsistencies in cutoff values used to define interleukin levels may have influenced the results. Finally, the derivation of certain data points from the included studies could have introduced potential discrepancies compared to the original data.

Conclusion

This meta-analysis demonstrates a strong association between interleukin levels and survival outcomes in BCa, suggesting that interleukin testing may serve as a valuable clinical tool for predicting patient outcomes and informing treatment decisions. Large-scale, multicenter prospective studies are recommended to further validate these findings.

Supplemental Information

Supplemental Information 1 PRISMA checklist.

Abbreviation

BCa Bladder cancer

IL-6 interleukin-6

IL-17 interleukin-17

PRISMA Preferred Reporting Items for Systematic Review and Meta-Analysis

MOOSE Meta-analysis of Observational Studies in Epidemiology

AMSTAR Assessing the methodological quality of systematic reviews

CNKI Chinese National Knowledge Infrastructure

DFS disease-free survival

OS overall survival

DSS disease-specific survival

DRS disease recurrence survival

RFS recurrence-free survival

PFS prognosis-free survival

CSS cancer-specific survival

HR hazard ratios

NOS Newcastle-Ottawa Quality Assessment Scale

RC radical cystectomy

TURBT transurethral resection of bladder tumour

Additional Information and Declarations

Competing Interests

The authors declare that they have no competing interests.

Author Contributions

Liang Liu conceived and designed the experiments, performed the experiments, analyzed the data, prepared figures and/or tables, authored or reviewed drafts of the article, and approved the final draft.

Jun-Hui He performed the experiments, authored or reviewed drafts of the article, and approved the final draft.

Yu Xiao performed the experiments, prepared figures and/or tables, authored or reviewed drafts of the article, and approved the final draft.

Dong Wei performed the experiments, prepared figures and/or tables, and approved the final draft.

Data Availability

The following information was supplied regarding data availability:

This is a systematic review/meta-analysis.

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
