# Peer review of "Prognostic impact of serum interleukin-6 and 17 level in patients with bladder cancer: a systematic review and meta-analysis"

_PeerJ, doi:10.7717/peerj.19385_

## Round 0.1 · original submission · Major Revisions

Please address all of the issues raised by the reviewers.

Reviewer 1 ·

Basic reporting

no comment

Experimental design

no comment

Validity of the findings

no comment

Additional comments

This manuscript (title: Prognostic impact of serum interleukin-6 and 17 level
in patients with bladder cancer: a systematic review and meta-analysis) aimed to evaluate the prognostic role of interleukin in bladder cancer.
Several critical issues warrant further attention:
The introduction could be improved by discussing how pathological variations in bladder cancer can impact treatment and prognosis.
The introduction mentions controversy regarding the research on interleukin-6 and 17; please provide examples.
I noticed that some of the studies included by the authors cover both NMIBC and MIBC. How did the authors handle the results from these studies during subgroup analysis?
For NMIBC, the mortality rate is relatively low. The results section primarily discusses OS and DFS; should the RFS for NMIBC be further compared? If not, please address this in the discussion section.

·

Basic reporting

Although this article demonstrates the systematicity and rigor of the research work in some aspects, there are significant shortcomings in multiple key areas that require significant revisions by the author to improve its academic quality and credibility. Firstly, although professional English is used throughout the article, the language expression in some paragraphs appears lengthy and not concise enough, which affects the fluency of reading. In addition, the use of some terms may not be accurate enough and requires further verification by the author to ensure professionalism and clarity. In terms of references, although the article provides relatively rich background information, the selection of some key citations seems not comprehensive enough to fully cover the latest developments and controversial points in the field. This may lead to a deviation in readers' understanding of the research background. Although the professional structure of the article basically meets the requirements of the journal, there are flaws in the organization and logical connection of some chapters. For example, the description of the methods and results section sometimes appears too brief and lacks sufficient details to support the validity of the conclusions. In addition, although the presentation of charts is intuitive, in some cases it fails to fully demonstrate the complexity and heterogeneity of the data. Regarding the sharing of raw data, the article failed to directly provide the raw dataset or access links, which severely limits readers' ability to independently verify and reanalyze research results. Against the backdrop of increasing emphasis on transparency in scientific research, this deficiency is unacceptable. In terms of self inclusion, although the article attempts to closely link the results with the hypotheses, the derivation process of some conclusions appears insufficiently rigorous and lacks sufficient evidence support. Especially for those findings that contradict existing literature, the author failed to provide convincing explanations or further evidence to strengthen their argument.

Experimental design

This study has the following shortcomings:
1. Literature inclusion criteria: Did the article fully consider key factors such as the design type, sample size, and follow-up time of the study when screening for inclusion? Is it possible that some high-quality research has been overlooked?
2. Heterogeneity treatment: In meta-analysis, there is significant heterogeneity between different studies (such as detection methods, cutoff values, etc. for IL-6 and IL-17). How does the article handle these heterogeneities? Have sufficient sensitivity and subgroup analyses been conducted to validate the stability of the results?
3. Bias assessment: Has the article conducted a bias risk assessment on the included studies? Have you considered factors such as publication bias and selection bias that may affect the results?
4. Data extraction and verification: Was the data extraction process completed by at least two independent researchers and cross checked? How to ensure the accuracy and completeness of extracted data?
5. Statistical analysis methods: Is the statistical analysis method used in the article (such as the selection of random effects models and fixed effects models) appropriate? Have all hypotheses been thoroughly tested?

Validity of the findings

I think there are the following questions that the author needs to answer:
1. Feasibility of clinical application: Although the article found the correlation between the levels of IL-6 and IL-17 and the prognosis of bladder cancer, are these findings enough to support the routine detection of these markers in clinical practice? What is the cost-benefit ratio?
2. Exploration of biological mechanisms: Although the article discusses the possible mechanisms of IL-6 and IL-17 in tumor occurrence and development, are these discussions sufficiently based on existing biological evidence? Is it necessary to conduct further basic research to verify these mechanisms?
3. Future research directions: Based on the current research results, does the article propose any forward-looking future research directions? For example, is it recommended to conduct large-scale, multicenter prospective studies to further validate these findings?

Additional comments

none

---

## Round 0.2 · Major Revisions

The authors should address all of the issues raised by Reviewer 3.

·

Basic reporting

The meta-analysis "Prognostic impact of serum interleukin-6 and 17 levels in bladder cancer patients: a systematic review and meta-analysis" offers an in-depth examination of how serum interleukin (IL-6 and IL-17) levels relate to survival outcomes in bladder cancer (BCa) patients.
The analysis is methodologically robust, structured effectively, and aligns with established guidelines. Nonetheless, several aspects should be refined or elaborated upon to improve clarity and overall impact.

Some sections in the introduction emphasize histologic variants and their prognostic significance, which, although pertinent, may divert attention from the main focus on interleukins.
The introduction briefly addresses the roles of IL-6 and IL-17 in other cancers but could further explain why these biomarkers are especially relevant to bladder cancer (e.g., specific pathways involved in BCa progression). Additionally, the conflicting findings of previous studies are noted, but they do not provide a clearer synthesis of the reasons behind these inconsistencies (e.g., differences in methodologies, patient cohorts, or statistical approaches).

Experimental design

The study follows PRISMA, MOOSE, and AMSTAR guidelines, ensuring high methodological rigor. The inclusion and exclusion criteria are clearly defined, enhancing clarity and reproducibility. However, incorporating observational studies, such as case-control and cohort designs, introduces inherent biases that need more explicit acknowledgment. Moreover, the handling of missing or inconsistent data, illustrated by variances in interleukin measurement methodologies, should be explained in greater detail.
Only seven studies were included in the meta-analysis, with even fewer studies (e.g., just two) examining IL-17. This limitation affects the statistical power and generalizability of the findings, especially regarding IL-17.
High heterogeneity (I² > 75%) was observed in several analyses, particularly for OS and DFS. While the authors used random-effects models to account for this, the heterogeneity suggests that the included studies may differ significantly regarding patient populations, methodologies, or other factors. This limits the ability to draw definitive conclusions.
The studies employed varying cut-off values to define "high" and "low" levels of IL-6 and IL-17, resulting in variability that complicates the interpretation of pooled results. Standardizing cut-off values across studies would enhance comparability.

Validity of the findings

While the authors propose that serum interleukin testing could be a valuable clinical tool, they fail to provide specific recommendations for practical implementation. For instance, how should clinicians interpret IL-6 and IL-17 levels alongside other prognostic factors?
Although the authors mentioned that elevated serum interleukin levels consistently predicted unfavorable prognosis irrespective of geographic origin, Hazard ratios for China and other countries are HR = 4.42; 95% CI: 1.90-10.30, and HR = 1.32; 95% CI: 1.23-1.41 respectively (Figure 5). How they came to the conclusion that there were no differences between the populations needs to be explained in more detail.

Additional comments

There is a minor point that needs to be corrected. Reference numbers are wrong in Table 2.

---

## Round 0.3 · accepted · Accept

The manuscript is ready for publication.

·

Basic reporting

No further comment.

Experimental design

No further comment.

Validity of the findings

No further comment.

Additional comments

Dear Editor,
I acknowledge that the authors have clarified the concerns I raised in my previous assessment. In my considered opinion, the article is now suitable for acceptance for publication in its current form.